# ViLBERT: Pretraining Task-Agnostic Visiolinguistic Representations for Vision-and-Language Tasks

**Jiasen Lu**[1], **Dhruv Batra**[1,3], **Devi Parikh**[1,3], **Stefan Lee**[1,2]

[1]Georgia Institute of Technology, [2]Oregon State University, [3]Facebook AI Research

## Abstract

We present ViLBERT (short for Vision-and-Language BERT), a model for learning task-agnostic joint representations of image content and natural language. We extend the popular BERT architecture to a multi-modal two-stream model, processing both visual and textual inputs in separate streams that interact through co-attentional transformer layers. We pretrain our model through two proxy tasks on the large, automatically collected Conceptual Captions dataset and then transfer it to multiple established vision-and-language tasks – visual question answering, visual commonsense reasoning, referring expressions, and caption-based image retrieval – by making only minor additions to the base architecture. We observe significant improvements across tasks compared to existing task-specific models – achieving state-of-the-art on all four tasks. Our work represents a shift away from learning groundings between vision and language only as part of task training and towards treating visual grounding as a pretrainable and transferable capability.

## 1 Introduction

> "... spend the summer linking a camera to a computer and getting the computer to describe what it saw."
>
> *Marvin Minsky on the goal of a 1966 undergraduate summer research project [1]*

Since this now famously ambitious summer project, steady progress has been made towards systems that can demonstrate their visual understanding by generating or responding to natural language in the context of images, videos, or even full 3D environments [2–8]. These approaches and corresponding tasks have come to be referred to under the common banner of 'vision-and-language'. However, despite the common need to align natural language and visual stimuli – *i.e.* to perform *visual grounding* – approaches for vision-and-language tasks lack a unified foundation to gain this capability. Instead, the dominant strategy is to start with separate language and vision models pretrained for other large-scale tasks and then learn grounding as part of task training – often resulting in myopic groundings that generalize poorly when paired visiolinguistic data is limited or biased [9, 10].

This *pretrain-then-transfer* learning approach to vision-and-language tasks follows naturally from its widespread use in both computer vision and natural language processing where it has become the de facto standard due to the ease-of-use and strong representational power of large, publicly-available models [11–14] trained on large-scale data sources [15–19]. In these domains, pretrained models can provide useful information for target tasks, *e.g.* dog breed-sensitive image features or a well-calibrated semantic distance between words. While visual and linguistic understandings like these are of course essential to vision-and-language tasks, equally important is how they relate to one another – *e.g.* a perfect visual representation of dog breeds is of little use if a downstream vision-and-language model fails to associate it with appropriate phrases like "beagle" or "shepherd". We are therefore interested in developing a common model for visual grounding that can learn these connections and leverage them on a wide array of vision-and-language tasks – *i.e.*, we seek to pretrain for visual grounding.

To learn these joint visual-linguistic representations, we look to recent successes in self-supervised learning which have captured rich semantic and structural information from large, unlabelled data sources by training models to perform so-called 'proxy' tasks. These proxy tasks leverage structure

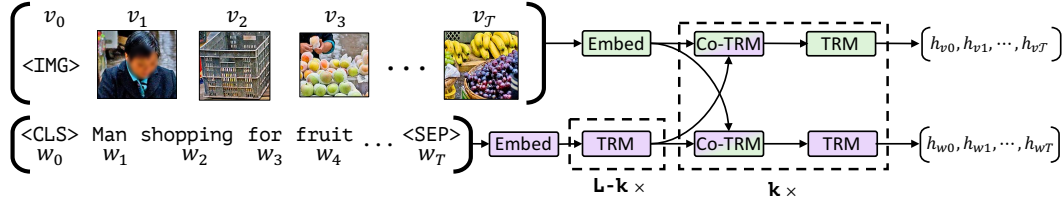

Figure 1: Our ViLBERT model consists of two parallel streams for visual (green) and linguistic (purple) processing that interact through novel co-attentional transformer layers. This structure allows for variable depths for each modality and enables sparse interaction through co-attention. Dashed boxes with multiplier subscripts denote repeated blocks of layers.

within the data to generate supervised tasks automatically (*e.g.* colorizing images [20] or reconstructing masked words in text [12]). While work within the vision community has shown increasing promise [21–23], the greatest impact of self-supervised learning so far is through language models like ELMo [13], BERT [12], and GPT [14] which have set new high-water marks on many NLP tasks. To learn visual grounding via a similar approach, we must identify a suitable data source where alignment between vision and language is available. In this work, we consider the recently released Conceptual Captions [24] dataset consisting of ∼3.3 million images with weakly-associated descriptive captions automatically collected from alt-text enabled images on the web.

We present a joint model for learning task-agnostic visual grounding from paired visiolinguistic data which we call Vision & Language BERT (ViLBERT for short). Our approach extends the recently developed BERT [12] language model to jointly reason about text and images. Our key technical innovation is introducing separate streams for vision and language processing that communicate through co-attentional transformer layers. This structure can accommodate the differing processing needs of each modality and provides interaction between modalities at varying representation depths. We demonstrate that this structure outperforms a single-stream unified model in our experiments.

In analogy to the training tasks in [12], we train our model on Conceptual Captions on two proxy tasks: predicting the semantics of masked words and image regions given the unmasked inputs, and predicting whether an image and text segment correspond. We apply our pretrained model as a base for four established vision-and-language tasks – visual question answering [3], visual commonsense reasoning [25], referring expressions [2], and caption-based image retrieval [26] – setting state-of-the-art on all four tasks. We find improvements of 2 to 10 percentage points across these tasks when compared to state-of-the-art task-specific baselines using separately pretrained vision and language models. Furthermore, our structure is simple to modify for each of these tasks – serving as a common foundation for visual grounding across multiple vision-and-language tasks.

## 2   Approach

In this section, we first briefly summarize the BERT language model (Sec. 2.1) and then describe how we extend it to jointly represent vision and language data (Sec. 2.2).

### 2.1   Preliminaries: Bidirectional Encoder Representations from Transformers (BERT)

The BERT model introduced by [12] is an attention-based bidirectional language model. When pretrained on a large language corpus, BERT has proven to be very effective for transfer learning to multiple natural language processing tasks.

The BERT model operates on sequences of word tokens $w_0, \ldots, w_T$. These tokens are mapped to learned encodings and passed through $L$ "encoder-style" transformer blocks [27] to produce final representations $h_0, \ldots, h_T$. Let $H^{(l)}$ be a matrix with rows $h_0^{(l)}, \ldots, h_T^{(l)}$ corresponding to the intermediate representations after the $l$-th layer. Abstracting some internal details found in [27], we depict the computation of a single encoder-style transformer block in Fig. 2a consisting of a multi-headed attention block followed by a small fully-connected network, both wrapped in residual adds. Note that the intermediate representation $H^{(l)}$ is used to compute three matrices – $Q$, $K$, and $V$ – corresponding to queries, keys, and values that drive the multi-headed attention block. Specifically, the dot-product similarity between queries and keys determines attentional distributions over value vectors. The resulting weight-averaged value vector forms the output of the attention block. As we describe later, we modify this query-conditioned key-value attention mechanism to develop a multi-modal co-attentional transformer module for ViLBERT (Fig. 2b).

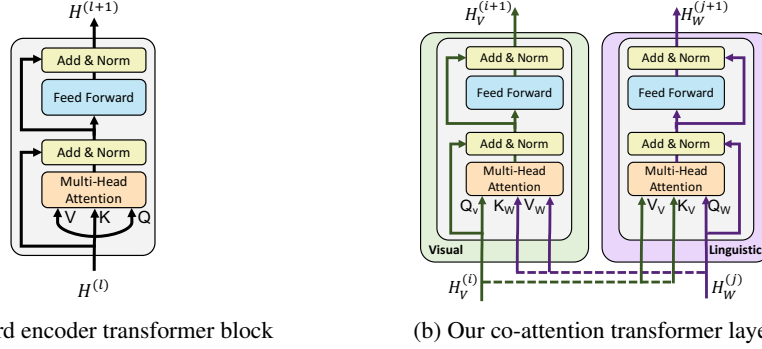

| (a) Standard encoder transformer block | (b) Our co-attention transformer layer |

Figure 2: We introduce a novel co-attention mechanism based on the transformer architecture. By exchanging key-value pairs in multi-headed attention, this structure enables vision-attended language features to be incorporated into visual representations (and vice versa).

**Text Representation.** BERT operates over sequences of discrete tokens comprised of vocabulary words and a small set of special tokens: SEP, CLS, and MASK. For a given token, the input representation is a sum of a token-specific learned embedding [28] and encodings for position (*i.e.* token's index in the sequence) and segment (*i.e.* index of the token's sentence if multiple exist).

**Training Tasks and Objectives.** The BERT model is trained end-to-end on a large language-corpus under two tasks: *masked language modelling* and *next sentence prediction*.

The masked language modelling task randomly divides input tokens into disjoint sets corresponding to masked $X_M$ and observed $X_O$ tokens (approximately 15% of tokens being masked). Masked tokens are replaced with a special MASK token 80% of the time, a random word 10%, and unaltered 10%. The BERT model is then trained to reconstruct these masked tokens given the observed set. Specifically, a linear layer is learned to map the final representations at each index (*e.g.* $h_i$) to a distribution over the vocabulary and the model is trained under a cross-entropy loss.

In next sentence prediction, the BERT model is passed two text segments $A$ and $B$ following the format {CLS, $w_{A1}, \ldots, w_{AT}$, SEP, $w_{B1}, \ldots, w_{B\mathcal{T}}$, SEP} and is trained to predict whether or not $B$ follows $A$ in the source text. Specifically, a linear layer operating on the final representation for the CLS token (*i.e.* $h_{\text{CLS}}$) is trained to minimize a binary cross-entropy loss on this label.

## 2.2 ViLBERT: Extending BERT to Jointly Represent Images and Text

Inspired by BERT's success at language modeling, we would like to develop analogous models and training tasks to learn joint representations of language and visual content from paired data. Specifically, we consider jointly representing static images and corresponding descriptive text.

One straightforward approach is to make minimal changes to BERT – simply discretizing the space of visual inputs via clustering, treat these visual 'tokens' exactly like text inputs, and start from a pretrained BERT model[1]. This architecture suffers from a number of drawbacks. First, initial clustering may result in discretization error and lose important visual details. Second, it treats inputs from both modalities identically, ignoring that they may need different levels of processing due to either their inherent complexity or the initial level of abstraction of their input representations. For instance, image regions may have weaker relations than words in a sentence and visual features are themselves often already the output of a very deep network. Finally, forcing the pretrained weights to accommodate the large set of additional visual 'tokens' may damage the learned BERT language model. Instead, we develop a two-stream architecture modelling each modality separately and then fusing them through a small set of attention-based interactions. This approach allows for variable network depth for each modality and enables cross-modal connections at different depths.

Our model which we call ViLBERT is shown in Fig. 1 and consists of two parallel BERT-style models operating over image regions and text segments. Each stream is a series of transformer blocks (TRM) and novel co-attentional transformer layers (Co-TRM) which we introduce to enable information exchange between modalities. Given an image $I$ represented as a set of region features $v_1, \ldots, v_{\mathcal{T}}$ and a text input $w_0, \ldots, w_T$, our model outputs final representations $h_{v0}, \ldots, h_{v\mathcal{T}}$ and $h_{w0}, \ldots, h_{wT}$. Notice that exchange between the two streams is restricted to be between specific

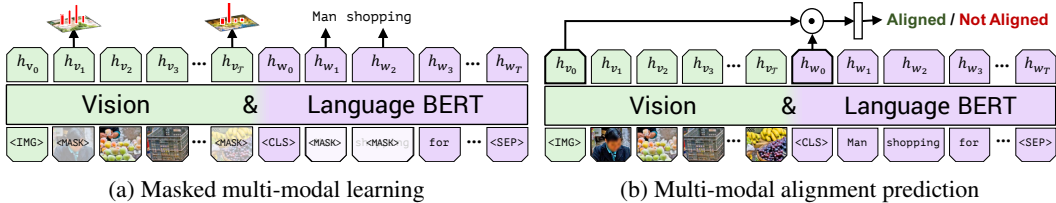

(a) Masked multi-modal learning       (b) Multi-modal alignment prediction

Figure 3: We train ViLBERT on the Conceptual Captions [24] dataset under two training tasks to learn visual grounding. In masked multi-modal learning, the model must reconstruct image region categories or words for masked inputs given the observed inputs. In multi-modal alignment prediction, the model must predict whether or not the caption describes the image content.

layers and that the text stream has significantly more processing before interacting with visual features – matching our intuitions that our chosen visual features are already fairly high-level and require limited context-aggregation compared to words in a sentence.

**Co-Attentional Transformer Layers.** We introduce a co-attentional transformer layer shown in Fig. 2b. Given intermediate visual and linguistic representations $H_V^{(i)}$ and $H_W^{(j)}$, the module computes query, key, and value matrices as in a standard transformer block. However, the keys and values from each modality are passed as input to the other modality's multi-headed attention block. Consequentially, the attention block produces attention-pooled features for each modality conditioned on the other – in effect performing image-conditioned language attention in the visual stream and language-conditioned image attention in the linguistic stream. The latter mimics common attention mechanisms found in vision-and-language models [30]. The rest of the transformer block proceeds as before, including a residual add with the initial representations – resulting in a multi-modal feature. In general, co-attention for vision-and-language is not a new idea (being first proposed in [31]) and concurrent work [32,33] has shown the effectiveness of similar co-attentional transformer structures on the visual question answering [3] task.

**Image Representations.** We generate image region features by extracting bounding boxes and their visual features from a pre-trained object detection network (see Sec. 3.1). Unlike words in text, image regions lack a natural ordering. we encode spatial location instead, constructing a 5-d vector from region position (normalized top-left and bottom-right coordinates) and the fraction of image area covered. This is then projected to match the dimension of the visual feature and they are summed.

We mark the beginning of an image region sequence with a special IMG token representing the entire image (i.e. mean-pooled visual features with a spatial encoding corresponding to the entire image).

**Training Tasks and Objectives.** In analogy to those described in the previous section, we consider two pretraining tasks: *masked multi-modal modelling* and *multi-modal alignment prediction*.

The masked multi-modal modelling task (shown in Fig. 3a) follows from the masked language modelling task in standard BERT – masking approximately 15% of both words and image region inputs and tasking the model with reconstructing them given the remaining inputs. Masked image regions have their image features zeroed out 90% of the time and are unaltered 10%. Masked text inputs are handled as in BERT. Rather than directly regressing the masked feature values, the model instead predicts a distribution over semantic classes for the corresponding image region. To supervise this, we take the output distribution for the region from the same pretrained detection model used in feature extraction. We train the model to minimize the KL divergence between these two distributions. This choice reflects the notion that language often only identifies high-level semantics of visual content and is unlikely to be able to reconstruct exact image features. Further, applying a regression loss could make it difficult to balance losses incurred by masked image and text inputs.

In the multi-modal alignment task (shown in Fig. 3b), the model is presented an image-text pair as $\{\texttt{IMG}, v_1, \ldots, v_{\mathcal{T}}, \texttt{CLS}, w_1, \ldots, w_T, \texttt{SEP}\}$ and must predict whether the image and text are aligned, *i.e.* whether the text describes the image. We take the outputs $h_{\texttt{IMG}}$ and $h_{\texttt{CLS}}$ as holistic representations of the visual and linguistic inputs. Borrowing another common structure from vision-and-language models, we compute the overall representation as an element-wise product between $h_{\texttt{IMG}}$ and $h_{\texttt{CLS}}$ and learn a linear layer to make the binary prediction whether the image and text are aligned. However, the Conceptual Captions [24] dataset only includes aligned image-caption pairs. To generate negatives for an image-caption pair, we randomly replace either the image or caption with another.

# 3 Experimental Settings

In this section, we describe how we train our model and provide overviews of the vision-and-language tasks to which we transfer the trained model.

## 3.1 Training ViLBERT

To train our full ViLBERT model, we apply the training tasks presented in Sec. 2.2 to the Conceptual Captions dataset [24]. Conceptual Captions is a collection of 3.3 million image-caption pairs automatically scraped from alt-text enabled web images. The automatic collection and sanitation process leaves some noise and the 'captions' are sometimes not human-like or short on details (*e.g.* "actors attend the premiere at festival"). However, it presents a huge diversity of visual content and serves as an excellent dataset for our purposes. Since some links had become broken by the time we downloaded the data, our model is trained with around 3.1 million image-caption pairs.

**Implementation Details.** We initialize the linguistic stream of our ViLBERT model with a BERT language model pretrained on the BookCorpus [17] and English Wikipedia. Specifically, we use the $\text{BERT}_{\text{BASE}}$ model [12] which has 12 layers of transformer blocks with each block having a hidden state size of 762 and 12 attention heads. We choose to use the $\text{BASE}$ model due to concerns over training time but find it likely the more powerful $\text{BERT}_{\text{LARGE}}$ model could further boost performance.

We use Faster R-CNN [31] (with ResNet-101 [11] backbone) pretrained on the Visual Genome dataset [16] (see [30] for details) to extract region features. We select regions where class detection probability exceeds a confidence threshold and keep between 10 to 36 high-scoring boxes. For each selected region $i$, $v_i$ is defined as the mean-pooled convolutional feature from that region. Transformer and co-attentional transformer blocks in the visual stream have hidden state size of 1024 and 8 attention heads.

We train on 8 TitanX GPUs with a total batch size of 512 for 10 epochs. We use the Adam optimizer with initial learning rates of 1e-4. We use a linear decay learning rate schedule with warm up to train the model. Both training task losses are weighed equally.

## 3.2 Vision-and-Language Transfer Tasks

We transfer our pretrained ViLBERT model to a set of four established vision-and-language tasks and one diagnostic task. We follow a fine-tuning strategy where we modify the pretrained base model to perform the new task and then train the entire model end-to-end. In all cases, the modification is trivial – typically amounting to learning a classification layer. This is in stark contrast to the significant efforts made within the community to develop specialized models for each of these tasks. We describe the problem, dataset, model modifications, and training objective for each task below.

**Visual Question Answering (VQA).** The VQA task requires answering natural language questions about images. We train and evaluate on the VQA 2.0 dataset [3] consisting of 1.1 million questions about COCO images [5] each with 10 answers. To fine-tune ViLBERT on VQA, we learn a two layer MLP on top of the element-wise product of the image and text representations $h_{\text{IMG}}$ and $h_{\text{CLS}}$, mapping this representation to 3,129 possible answers. As in [30], we treat VQA as a multi-label classification task – assigning a soft target score to each answer based on its relevancy to the 10 human answer responses. We then train with a binary cross-entropy loss on the soft target scores using a batch size of 256 over a maximum of 20 epochs. We use the Adam optimizer with an initial learning rate of 4e-5. At inference, we simply take a softmax.

**Visual Commonsense Reasoning (VCR).** Given an image, the VCR task presents two problems – visual question answering (Q→A) and answer justification (QA→R) – both being posed as multiple-choice problems. The holistic setting (Q→AR) requires both the chosen answer and then the chosen rationale to be correct. The Visual Commonsense Reasoning (VCR) dataset consists of 290k multiple choice QA problems derived from 110k movie scenes. Different from the VQA dataset, VCR integrates object tags into the language providing direct grounding supervision and explicitly excludes referring expressions. To finetune on this task, we concatenate the question and each possible response to form four different text inputs and pass each through ViLBERT along with the image. We learn a linear layer on top of the post-elementwise product representation to predict a score for each pair. The final prediction is a softmax over these four scores and is trained under a cross-entropy loss over 20 epochs with a batch size of 64 and initial learning rate of 2e-5.

**Grounding Referring Expressions.** The referring expression task is to localize an image region given a natural language reference. We train and evaluate on the RefCOCO+ dataset [32]. A common approach to this task is to rerank a set of image region proposals given the referring expression.

Thus we directly use the bounding box proposals provided by [33], which use a Mask R-CNN [34] pretrained on the COCO dataset. For fine-tuning, we pass the final representation $h_{v_i}$ for each image region $i$ into a learned linear layer to predict a matching score. We label each proposal box by computing the IoU with the ground truth box and thresholding at 0.5. We train with a binary cross-entropy loss for a maximum of 20 epochs with a batch size of 256 and an initial learning rate of 4e-5. At inference, we use the highest scoring region as the prediction.

**Caption-Based Image Retrieval.** Caption-based image retrieval is the task of identifying an image from a pool given a caption describing its content. We train and evaluate on the Flickr30k dataset [26] consisting of 31,000 images from Flickr with five captions each. Following the splits in [35], we use 1,000 images for validation and test each and train on the rest. These captions are well-grounded in and descriptive of the visual content and are qualitatively different than the automatically collected Conceptual Captions. We train in a 4-way multiple-choice setting by randomly sampling three distractors for each image-caption pair – substituting a random caption, a random image, or a hard negative from among the 100 nearest neighbors of the target image. We compute the alignment score (as in alignment prediction pretraining) for each and apply a softmax. We train this model under a cross-entropy loss to select the true image-caption pair for 20 epochs with a batch size of 64 and an initial learning rate of 2e-5. At inference, we score each caption-image pair in the test set and then sort. For efficiency, we cache the linguistic stream representation before the first Co-TRM layer – effectively freezing the linguistic representation before fusion.

**'Zero-shot' Caption-Based Image Retrieval.** The previous tasks are all transfer tasks that include dataset specific fine-tuning. In this 'zero-shot' task, we directly apply the pretrained the multi-modal alignment prediction mechanism to caption-based image retrieval in Flickr30k [26] *without fine-tuning* (thus the description as 'zero-shot'). The goal of this task is to demonstrate that the pretraining has developed the ability to ground text and that this can generalize to visual and linguistic variation without any task specific fine-tuning. We directly use the ViLBERT model trained on Conceptual Captions dataset described in Sec. 3.1. We use the alignment prediction objective as a scoring function and test on the same split as the caption-based image retrieval task described above.

## 4   Results and Analysis

**Baselines.** We compare our pretrained ViLBERT model against two ablative baselines:

– **Single-Stream** consisting of a single BERT architecture that processes both modality inputs through the same set of transformer blocks – sharing parameters and processing stacks for both visual and linguistic inputs. Like [29], this model avoids making changes to the BERT architecture, resulting in significantly deeper visual processing and earlier interaction between modalities than in our model. The model is initialized with BERT$_{\text{BASE}}$ and trained identically to our full model. We compare to this baseline to establish the impact of our two-stream architecture. As both streams interact throughout, we cannot cache any representations for efficiency. As such, we do not evaluate this baseline on image retrieval and zero-shot image retrieval due to high computational cost.

– **ViLBERT**[†] which is a ViLBERT architecture that has *not undergone our pretraining tasks*. Notably, it does still have BERT initilization for the linguistic stream and represents image regions with the same Faster R-CNN model as the full ViLBERT model. We compare to this baseline to isolate gains over task-specific baseline models that might be due to our architecture, language initialization, or visual features as opposed to our pretraining process on Conceptual Captions .

For both baselines and our model, we finetune the transfer tasks as described in the previous section.

**Task-Specific Baselines.** To put our results in context, we present published results of problem-specific methods that are to our knowledge state-of-the-art in each task: DFAF [36] for VQA, R2C [25] for VCR, MAttNet [33] for RefCOCO+, and SCAN [35] for caption-based image retrieval.

**Results.** Tab. 1 shows results across all transfer tasks and we highlight key findings below:

– **Our architecture improves performance over a single-stream model.** We observe improvements across tasks for ViLBERT over the single-stream baseline for both pretrained (Single-Stream vs. ViLBERT) and non-pretrained (Single-Stream[†] vs. ViLBERT[†]). Most significant gains are observed for VQA and RefCOCO+.

– **Our pretraining tasks result in improved visiolinguistic representations.** Our models further improve by between 2% and 13% across tasks when using a ViLBERT model that has been

Table 1: Transfer task results for our ViLBERT model compared with existing state-of-the-art and sensible architectural ablations. † indicates models without pretraining on Conceptual Captions. For VCR and VQA which have private test sets, we report test results (in parentheses) only for our full model. Our full ViLBERT model outperforms task-specific state-of-the-art models across all tasks.

| | Method | VQA [3] | VCR [25] | | | RefCOCO+ [32] | | | Image Retrieval [26] | | | ZS Image Retrieval | | |
|---|---|---|---|---|---|---|---|---|---|---|---|---|---|---|
| | | test-dev (test-std) | Q→A | QA→R | Q→AR | val | testA | testB | R1 | R5 | R10 | R1 | R5 | R10 |
| SOTA | DFAF [36] | 70.22 (70.34) | - | - | - | - | - | - | - | - | - | - | - | - |
| | R2C [25] | - | 63.8 (65.1) | 67.2 (67.3) | 43.1 (44.0) | - | - | - | - | - | - | - | - | - |
| | MAttNet [33] | - | - | - | - | 65.33 | 71.62 | 56.02 | - | - | - | - | - | - |
| | SCAN [35] | - | - | - | - | - | - | - | 48.60 | 77.70 | 85.20 | - | - | - |
| Ours | Single-Stream† | 65.90 | 68.15 | 68.89 | 47.27 | 65.64 | 72.02 | 56.04 | - | - | - | - | - | - |
| | Single-Stream | 68.85 | 71.09 | 73.93 | 52.73 | 69.21 | 75.32 | 61.02 | - | - | - | - | - | - |
| | ViLBERT† | 68.93 | 69.26 | 71.01 | 49.48 | 68.61 | 75.97 | 58.44 | 45.50 | 76.78 | 85.02 | 0.00 | 0.00 | 0.00 |
| | ViLBERT | **70.55 (70.92)** | **72.42 (73.3)** | **74.47 (74.6)** | **54.04 (54.8)** | **72.34** | **78.52** | **62.61** | **58.20** | **84.90** | **91.52** | **31.86** | **61.12** | **72.80** |

pretrained under our proxy tasks (ViLBERT vs ViLBERT† ). We also observe improvements on Single-Stream which verifies our proxy tasks can generalize to different model architectures.

– **Finetuning from ViLBERT is a powerful strategy for vision-and-language tasks.** With a single base architecture, our transfer task performance exceeds state-of-the-art task-specific models for all four established tasks. We set state-of-the-art for VCR, RefCOCO+ and image retrieval by significant margins (7-10 percentage points improvement). Further, extending to these tasks was simple – requiring the addition of a single classifier for each task.

Overall, these results demonstrate that our ViLBERT model is able to learn important visual-linguistic relationships that can be exploited by downstream tasks.

**Effect of Visual Stream Depth.** In Tab. 2 we compare the results transferring from ViLBERT models of varying depths. We consider depth with respect to the number of repeated CO-TRM→TRM blocks (shown in a dashed box in Fig. 1) in our model. We find that VQA and Image Retrieval tasks benefit from greater depth - performance increases monotonically until a layer depth of 6. Likewise, zero-shot image retrieval continues making significant gains as depth increases. In contrast, VCR and RefCOCO+ seem to benefit from shallower models.

**Benefits of Large Training Sets.** We also studied the impact of the size of the pretraining dataset. For this experiment, we take random subsets of 25% and 50% from the conceptual caption dataset, and pretrain and finetune ViLBERT using the same setup as above. We can see that the accuracy grows monotonically as the amount of data increases, which suggests that ViLBERT may benefit from even more pretraining data.

**What does ViLBERT learn during pretraining?** To get a sense for what ViLBERT learns during Conceptual Caption pretraining, we look at zero-shot caption-based image retreival and some qualitative examples. While zero-shot performance (Tab. 1, right) is significantly lower than the fine-tuned model (31.86 vs 58.20 R1) it performs reasonably without having seen a Flickr30k image or caption (31.86 vs 48.60 R1 for prior SOTA) – indicating that ViLBERT has learned a semantically meaningful alignment between vision and language during pretraining.

## 5 Related Work

**Self-Supervised Learning.** There has been substantial recent interest in both vision [37–42] and language around self-supervised representation learning. In this paradigm, deep models are trained

Table 2: Ablation study of the depth of our model with respect to the number of Co-TRM→TRM blocks (shown in a dashed box in Fig. 1). We find that different tasks perform better at different network depths – implying they may need more or less context aggregation.

| | VQA [3] | VCR [25] | | | RefCOCO+ [32] | | | Image Retrieval [26] | | | ZS Image Retrieval [26] | | |
|---|---|---|---|---|---|---|---|---|---|---|---|---|---|
| Method | test-dev | Q→A | QA→R | Q→AR | val | testA | testB | R1 | R5 | R10 | R1 | R5 | R10 |
| ViLBERT (2-layer) | 69.92 | 72.44 | **74.80** | **54.40** | 71.74 | **78.61** | 62.28 | 55.68 | 84.26 | 90.56 | 26.14 | 56.04 | 68.80 |
| ViLBERT (4-layer) | 70.22 | **72.45** | 74.00 | 53.82 | 72.07 | 78.53 | **63.14** | 55.38 | 84.10 | 90.62 | 26.28 | 54.34 | 66.08 |
| ViLBERT (6-layer) | **70.55** | 72.42 | 74.47 | 54.04 | **72.34** | 78.52 | 62.61 | 58.20 | 84.90 | **91.52** | 31.86 | 61.12 | 72.80 |
| ViLBERT (8-layer) | 70.47 | 72.33 | 74.15 | 53.79 | 71.66 | 78.29 | 62.43 | **58.78** | **85.60** | 91.42 | **32.80** | **63.38** | **74.62** |

Table 3: Transfer task results for ViLBERT as a function of the percentage of the Conceptual Captions dataset used during pre-training. We see monotonic gains as the pretraining dataset size grows.

| Method | VQA [3] | VCR [25] | | | RefCOCO+ [32] | | | Image Retrieval [26] | | | ZS Image Retrieval [26] | | |
|---|---|---|---|---|---|---|---|---|---|---|---|---|---|
| | test-dev | Q→A | QA→R | Q→AR | val | testA | testB | R1 | R5 | R10 | R1 | R5 | R10 |
| ViLBERT (0 %) | 68.93 | 69.26 | 71.01 | 49.48 | 68.61 | 75.97 | 58.44 | 45.50 | 76.78 | 85.02 | 0.00 | 0.00 | 0.00 |
| ViLBERT (25 %) | 69.82 | 71.61 | 73.00 | 52.66 | 69.90 | 76.83 | 60.99 | 53.08 | 80.80 | 88.52 | 20.40 | 48.54 | 62.06 |
| ViLBERT (50 %) | 70.30 | 71.88 | 73.60 | 53.03 | 71.16 | 77.35 | 61.57 | 54.84 | 83.62 | 90.10 | 26.76 | 56.26 | 68.80 |
| ViLBERT (100 %) | **70.55** | **72.42** | **74.47** | **54.04** | **72.34** | **78.52** | **62.61** | **58.20** | **84.90** | **91.52** | **31.86** | **61.12** | **72.80** |

for tasks where regularities in existing data can be turned into supervision automatically. While there has been progress on the vision side, self-supervised image representations still lag behind those from models trained under image classification tasks. Self-supervised language models on the other hand have resulted in significant improvements over prior work [12–14, 43]. In this work, we develop a model and proxy tasks for learning joint visual-linguistic representations – extending the popular BERT [12] model.

**Vision-and-Language.** While we address many vision-and-language tasks in Sec. 3.2, we do miss some families of tasks including visually grounded dialog [4, 44], embodied tasks like question answering [7] and instruction following [8], and text generation tasks like image and video captioning [5]. These tasks may also benefit from a self-supervised approach similar to what we have presented. There are open questions on how to incorporate long sequences of images and text found in dialog, embodied tasks, and video processing. Further, it is unclear how to effectively decode output text from our bidirectional model as existing greedy decoders like beam-search do not apply.

**Self-Supervised Learning for Vision-And-Language.** Most related to our approach is concurrent work on learning joint representations between video and language [29]. In this work, self-supervised tasks paralleling our own are derived from cooking videos paired with text-to-speech transcribed audio. They present a unified BERT architecture for both the visual and linguistic inputs similar to the Single-Stream baseline we consider here. They apply the learned model to two tasks on cooking videos: zero-shot activity recognition and blank-filling on audio transcripts. In contrast, we learn representations of images and descriptive text on a wide range of images from the web and focus extensively on transfer learning from this model for well-established vision-and-language tasks.

**Recent works on Vision-And-Language pre-training.** Since our paper released on arXiv, a few other useful preprints have recently been released on similar vision-and-language cross-modality pre-training directions. LXMERT [45] uses a more specific design for the cross-modality model. Instead of using webly supervised Conceptual Caption [24] dataset, LXMERT uses in-domain datasets (*i.e.* COCO [5] and Visual Genome [16]) for pre-training. VisualBERT [46] directly extend BERT [12] for vision and language domain. VisualBERT uses both out-of-domain and in-domain dataset for pre-training and applies MLM object only on the language side. Unicoder [47] focuses exclusively on image caption retrieval tasks with online hardest negative mining. More recent preprints including VLBERT [48], Unified VLP [49] and UNITER [49] also show promising improvements in this research direction of joint visio-linguistic pretraining.

# 6 Conclusion

We develop a joint model for image content and text and pretrain it on a large, automatically-collected dataset to learn visual grounding. Our ViLBERT model introduces a novel two-stream architecture with co-attentional transformer blocks that outperforms sensible ablations and exceeds state-of-the-art when transferred to multiple established vision-and-language tasks. Furthermore, transferring our model to these tasks is simple and easy to implement – requiring only the addition of a classifier for each task we examined here. We consider extensions of our model to other vision-and-language tasks (including those requiring generation) as well as multi-task learning as exciting future work.

**Acknowledgement.** The Georgia Tech effort was supported in part by NSF, AFRL, DARPA, ONR YIPs, ARO PECASE. The views and conclusions contained herein are those of the authors and should not be interpreted as necessarily representing the official policies or endorsements, either expressed or implied, of the U.S. Government, or any sponsor.

## Footnotes

[1]Concurrent work [29] modelling language and video sequences takes this approach. See Sec. 5.

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
