[Reviews · NeurIPS 2019]

Reviewer 1



The methods appear to be new, but they are mostly a collection of pretrained components with minor novelty linking the visual and linguistic components. The submission appears to be technically sound, and the experimental results seem to validate the idea put forth. The analysis is lacking; all the results point to one thing, the effectiveness of the pretraining method for transfer learning. That is the one point, and it seems to be demonstrated, but there is little further discussion. The submission is clear enough, and it is organized well enough to make an easy read. The fact that code is available aids in reproducibility where one might be in doubt about reproducing form the paper alone. The results are not particularly surprising, and they do not seem particularly revolutionary. Rather they seem like a reasonable next step for extending BERT. This is not to say the results are not valuable, only to properly scope the importance of this singular work given that much similar work is likely. Others will likely make use of this work and refer to it in the multimodal setting.

Reviewer 2



I think that this paper is a solid extension of masked language model pre-training to image-and-text (e.g., captioning) tasks. It defines two novel but intuitive pre-training tasks for this scenario: (i) predicting the semantic class of masked image regions given the surrounding image regions (from the same image) and the corresponding text, (ii) predicting whether image and text pairs are aligned. They demonstrate significant improvements over both the previous SOTA and the strong baseline of simply using a pre-trained text-only BERT model. They also show that having two encoders (with different parameters), one for images and one for text, is superior to a joint encoder. I would have liked to have seen more ablation of the pre-training tasks, since I think that this is more interesting than the model depth ablation that the authors performed. I think that the biggest weakness of the paper is that all of the experiments were pre-trained on Conceptual Captions and then evaluated on other image captioning (or closely related tasks). So effectively this can also be thought of as transfer learning from a large captioning data set to a small one, which is well-known to work. It would have been nice to see what the results could be with just image and text data without correspondences, as an additional ablation. The paper does have significant overlap with VideoBERT, but since the work is concurrent I don't think it's fair to penalize this paper because VideoBERT was uploaded to Arxiv first, so I did not factor that in.

Reviewer 3



Strengths: - Reusable and task agnostic visual-linguistic representations are very interesting approaches to tackle visual grounding problem. - The authors adapted the commonly known BERT to a new multimodal task.

[Author Response · NeurIPS 2019]

We thank the reviewers for the thoughtful feedback! We are encouraged that all voted to accept, finding the paper clear / well-organized [R1]; our approach "very interesting" [R3] and novel [R2 R3]; our results significant and well-demonstrated [R1 R2]; and likely to be built on by the community [R1]. We are pleased they recognized the value of transferring visio-linguistic pretraining [R1 R2 R3] and the demonstrated benefits of our co-attentional two-stream model over a direct extension of BERT [R2 R3]. We respond to select comments below but will address all feedback.

**Improved performance.** After submission, refined LR schedules raised performance across all tasks – passing the recent VQA challenge winner, setting a new SoTA. Will update paper with details and release the codebase if accepted.

[CLS] a sunny day with a cottage and trees . [SEP]

[CLS] a squirrel on a bench in a park [SEP]

**[R1] Visualization of coattention over multimodal inputs.** Visualizing BERT-style models is an open research area; we applied the method of [Vig. arXiv 2019] and observed some trends – providing representative examples on the left. **[Top]** We show attention for each layer (rows) and head (cols) with attention focus (colored lines) shown going from source to target (left-to-right). text→image co-attention tends to be better grounded in early layers before converging to a set of somewhat arbitrary regions as depth increases – see attention focus concentrate more over layers. In contrast, image→text co-attention often focuses on the high-level sentence representation in the SEP token already developed on the text side early on, but spreads out somewhat later. **[Bottom]** We also show the most attended patch for each attention head for each word in the first layer for an example. Many heads focus on a small set of "default" patches (faded for clarity); but, the noun phrases surrounding "squirrel" and "bench" focus more on relevant regions.

**[R1] Additional result analysis.** We investigate the RefCOCO+ task. For each noun occurring in a referring expression, we counted the number of instances where ViLBERT (full) succeeded and ViLBERT (w/o pretrain) fails (and vice versa). The wordcloud on the right shows those nouns with the highest performance delta. We will perform more task specific task in supplementary.

**[R2] All tasks are "image captioning (or closely related)" so this is "effectively transfer learning from a large captioning dataset to a small one."** We respectfully disagree. Due to its automatic collection from the web, Conceptual Captions (CC) is fairly distinct from curated vision-and-language datasets (examples right). Even for the closely-related caption-based image retrieval task, it was not obvious to us that this weakly-aligned web data would help. Further, our other transfer tasks differ significantly from CC. VQA and VCR both ask grounded questions like "Is there something to cut the vegetables with?" (VQA) This is not caption-like and requires reasoning (knives cut) and grounding. VCR extends to answer justifications like "[Person3] is delivering food to the table, and she might not know whose order is whose" that often refer to actions and intentions of individuals. Referring expressions focus on aligning small image regions with short focused text like "guy in yellow dribbling ball" – both being quite different from whole-image descriptive captioning. *However, a common need for visual grounding underpins these tasks and is precisely what we target with our pretraining strategy.*

**A.** exercises during a training session
**B.** deer while walking a dog
**C.** diagram of a modern sowing machine
**D.** fishing boat returning to port in winter in mid afternoon, a frigid breeze giving lie to the warm glowing light

**[R2] Additional pretraining ablations.** Great suggestions! We report separate image-text pretraining (**w/o corr** – masking loss only and zeroed co-attn), without alignment loss (**w/o align**), and without masking loss (**w/o mask**) ablations to the right (only two tasks due to time). All ablations degrade performance – especially **w/o mask** which struggles to train downstream tasks. These ablations are valuable and will be added to the paper.

| Method | VQA | RefCOCO+ |
|---|---|---|
| full | **66.59** | **70.38** |
| w/o corr | 64.85 | 68.04 |
| w/o align | 64.61 | 68.49 |
| w/o mask | 42.43 | 10.00 |

**[R3] Given the use of Conceptual Captions (CC), are the comparisons to baselines fair?** We believe these comparisons are fair. We agree that CC is a large, additional data source; however, being able to leverage this additional data for a diverse range of vision and language tasks is precisely our contribution! Existing approaches to vision and language tasks are simply not designed to do so – for instance, it is unclear how to train a standard VQA model like BAN with CC captioning data. Arguing from analogy, the widespread transfer of deep models pretrained on ImageNet also leveraged more data during pretraining; however, we do not find it unfair to pre-deep learning approaches that were not equipped to leverage that data. Finally, note that unlike ImageNet, CC is webly supervised, and did not involve expensive human annotation. We acknowledge that in caption-based image retrieval, CC data could have been used to pretrain existing work for a more direct architectural comparison – we will address.

**[R3] "If I understood correctly, [the w/o pretrain model] does not use any visual features.** That is not the case. Like all our models, the "w/o pretrain" model is initialized from a trained visual feature extractor (Faster RCNN) and language model (BERT). We use "w/o pretrain" to note that the model has not undergone our visio-linguistic pretraining on the CC dataset (L264). To reduce confusion, we will use "w/o grounding pretraining" and clarify relevant sentences.

[Meta-Review · NeurIPS 2019]

After rebuttal the reviewers recommend acceptance (8,7,6). The authors are encouraged to take into account reviewers' suggestions when preparing the camera ready.